Comparative transcriptome analysis of papilla and skin in the sea cucumber, Apostichopus japonicus

Zhou Xiaoxu 1 2
Cui Jun 2
Liu Shikai 3
Kong Derong 2
Sun He 2
Gu Chenlei 2
Wang Hongdi 2
Qiu Xuemei 1 2
Chang Yaqing 1 2
Liu Zhanjiang 3
Wang Xiuli 1 2 xiuliwang417@sina.com
1 Key Laboratory of Mariculture & Stock Enhancement in North China’s Sea, Ministry of Agriculture, Dalian Ocean University , Dalian , China
2 College of Fisheries and Life Science, Dalian Ocean University , Dalian , China
3 The Fish Molecular Genetics and Biotechnology Laboratory, Aquatic Genomics Unit, School of Fisheries, Aquaculture and Aquatic Sciences and Program of Cell and Molecular Biosciences, Auburn University , Auburn, Alabama , United States
Ng Tzi Bun
Electronic publication date: 2016 Mar 7
Publication date: 2016
Volume: 4
Electronic Location ID: e1779
Received 2015 Nov 12; Accepted 2016 Feb 17
Copyright: © 2016 Zhou et al.
Copyright year: 2016
Copyright holder: Zhou et al.
License: This is an open access article distributed under the terms of the Creative Commons Attribution License, which permits unrestricted use, distribution, reproduction and adaptation in any medium and for any purpose provided that it is properly attributed. For attribution, the original author(s), title, publication source (PeerJ) and either DOI or URL of the article must be cited.
License URL: https://creativecommons.org/licenses/by/4.0/

Keywords: Comparative transcriptome, High-throughput sequencing, Gene expression, Papilla, Skin, Sea cucumber (Apostichopus japonicus)

Funding: State 863 High-Technology R & D Project of China 2012AA10A412 Program for Liaoning Excellent Talents in University, China LR2014022 This project was supported by the State 863 High-Technology R & D Project of China (No. 2012AA10A412) and the Program for Liaoning Excellent Talents in University, China (No. LR2014022). The funders had no role in study design, data collection and analysis, decision to publish, or preparation of the manuscript.

==============================
Papilla and skin are two important organs of the sea cucumber. Both tissues have ectodermic origin, but they are morphologically and functionally very different. In the present study, we performed comparative transcriptome analysis of the papilla and skin from the sea cucumber (Apostichopus japonicus) in order to identify and characterize gene expression profiles by using RNA-Seq technology. We generated 30.6 and 36.4 million clean reads from the papilla and skin and de novo assembled in 156,501 transcripts. The Gene Ontology (GO) analysis indicated that cell part, metabolic process and catalytic activity were the most abundant GO category in cell component, biological process and molecular funcation, respectively. Comparative transcriptome analysis between the papilla and skin allowed the identification of 1,059 differentially expressed genes, of which 739 genes were expressed at higher levels in papilla, while 320 were expressed at higher levels in skin. In addition, 236 differentially expressed unigenes were not annotated with any database, 160 of which were apparently expressed at higher levels in papilla, 76 were expressed at higher levels in skin. We identified a total of 288 papilla-specific genes, 171 skin-specific genes and 600 co-expressed genes. Also, 40 genes in papilla-specific were not annotated with any database, 2 in skin-specific. Development-related genes were also enriched, such as fibroblast growth factor, transforming growth factor-β, collagen-α2 and Integrin-α2, which may be related to the formation of the papilla and skin in sea cucumber. Further pathway analysis identified ten KEGG pathways that were differently enriched between the papilla and skin. The findings on expression profiles between two key organs of the sea cucumber should be valuable to reveal molecular mechanisms involved in the development of organs that are related but with morphological differences in the sea cucumber.

Introduction

The sea cucumbers group (Echinodermata, Holothuroidea) comprises approximately 1,250 species (Du et al., 2012). Sea cucumbers are mostly processed into a dry product called trepang, bêche-de-mer or hai-san, which is widely recognized as a delicate food with a medicinal effect for human consumption. Sea cucumbers have been harvested for commercial use for a thousand years, and they are now widely cultured in more than 70 countries (Steven et al., 2012). The sea cucumber Apostichopus japonicus (Holothuroidea, Aspidochirotida) is intensively cultured in many East Asian countries and is naturally found along the coasts of China, Japan, Korea and Russia of Northeast Asia (Sloan, 1984; Chang et al., 2009). It is intensively cultured as an important aquaculture species in many countries of East Asia.

The pentamerous radial symmetry is considered as one of the characteristics of echinodermata. In sea cucumber, pentamerous symmetry is usually determined based on the presence of five meridional ambulacra bearing podia (Steven et al., 2012). Papillae represent the podia on the dorsal surface, and generally have no locomotive function. With A. japonicus, fleshy and conical papillae, with a sensory spina at its apex, are present in two loose rows on the dorsal surface and two rows at the lateral margins of the ventral surface (Steven et al., 2012). Previous studies have investigated the morphological characteristics of papilla in the A. japonicus (Vanden-Spiegel et al., 1995; Chang et al., 2011; Steven et al., 2012). In the papillae, the ciliated cells and histamine-like immunoreactivity neurons are in contact with the nerve plexus (Hyman, 1955; Luke et al., 2012). Therefore, the dorsal papillae have long been associated to a sensory role, which may involve chemoreception and mechanoreception (Vanden-Spiegel et al., 1995).

The thicker body wall of A. japonicus consists of a thin cuticle over the epidermis and a thick dermis underneath. The cuticle and epidermis as the outer tissues of the A. japonicus are represented by skin (Steven et al., 2012). The skin forms a protective barrier, forming the first line of defence against the environment. Previous studies have been conducted on skin, with the main focus on the intrinsic mechanisms underlying immune response to skin ulceration and peristome tumescence (Liu et al., 2010; Zhang et al., 2013).

The papillae are closely associated with the skin in sea cucumber. Both organs are mainly composed of collagen (up to 70%), and are the major component of the body wall. Moreover, the papillae and skin are formed by similar elements and homologous cell types, such as keratinocytes, epidermis and dermis, all derived from the ectoderm (Chang et al., 2004; Lowdon et al., 2014). Despite the common embryonic origin of the two organs, they exhibit clear morphological differences and play distinct functions. The molecular mechanisms underlying differentiation between the papilla and skin remain largely unknown. The lack of reference genome and the limited genetic resources of A. japonicus represent a major obstacle to better understand the function of these two organs.

In this study, we conducted RNA-Seq of these two organs to determine global changes in gene expression between the papilla and skin in the A. japonicus. RNA-Seq technology has been widely used for the generation of genetic resources in echinoderms (Anisimov, 2008; Wang, Gerstein & Snyder, 2009). Recently, several RNA-Seq based transcriptome analyses have been conducted in the A. japonicus, including studies on histology (Sun et al., 2011; Sun et al., 2013), immunology (Li et al., 2012), physiology (Zhao et al., 2014a; Zhao et al., 2014b), embryonic development and gene marker discovery (Du et al., 2012; Zhou et al., 2014). The first transcriptome sequencing of the A. japonicus intestine and body wall was performed by Sun et al. (2011). Thereafter, the global dynamic changes during all stages of intestine regeneration were further investigated (Sun et al., 2013). To identify candidate transcripts potentially involved in aestivation and generate a wide coverage of transcripts involved in a broad range of biological processes, eight cDNA libraries were constructed and sequenced by Du et al. (2012). Immune-related genes and pathways in response to pathogen infection were identified (Zhou et al., 2014; Gao et al., 2015). Moreover, many physiological networks were identified and characterized in the A. japonicus on the basis of transcriptomic resources (Wang et al., 2015; Yang et al., 2015).

Here, in this work, we report comparative transcriptome analysis of the papilla and skin. A relatively large number of genes that displayed distinct expression profiles between the papillae and skin were identified. Further enrichment analysis identified pathways such as tight junction and p53 signaling pathway could be involved in the development of the papilla and skin. This work provided the essential genomic resources for further investigations into the molecular interactions and multiple biological process of appendages such as the papilla and skin in the A. japonicus.

Materials and Methods

Sample collection

A total of 45 sea cucumbers (average weight of 25 g) provided by the Key Laboratory of Mariculture in North China (Dalian, Liaoning) were used in the present study. In order to have a good reference transcriptome, the skin around the papillae, papilla and tube foot tissues were collected for RNA-Seq. We randomly group these 45 sea cucumbers into three groups as replicates. Within each group, ∼1 g tissue was dissected from each individual, respectively. Tissues collected from each group were of every individual were pooled (one pool per tissue) and placed in 2 ml of RNAlater®Solution (Ambion, Carlsbad, CA, USA) for overnight at 4 °C followed by transferring to −80 °C until RNA extraction.

RNA-Seq

Total RNA was extracted from the pooled samples using the TRIzol Reagent (Invitrogen, Carlsbad, CA, USA) following the manufacturer’s recommendations. The quantity and integrity of total RNA were assessed using an Agilent 2100 Bioanalyzer and 1% agarose gel electrophoresis. High quality RNA was used for the construction of cDNA. Library construction and sequencing was performed in the Biomarker Biotechnology Corporation (Beijing, China). Paired-end sequencing was conducted on an Illumina HiSeq 2500 platform to generate 125 bp Paired-End (PE) reads.

Transcriptome assembly and annotation

Low quality reads and adaptors were trimmed before assembly. Trimed reads were de novo assembled by Trinity software using default parameters (Grabherr et al., 2011) and used as a reference for gene expression analysis. Transcirptome was annotated using Basic Local Alignment Search Tool (BLAST) searches against the NCBI Non-Redundant (NR) database, Swiss-Prot, KEGG (the Kyoto Encyclopedia of Genes and Genomes) and GO (Gene Ontology), Clusters of Orthologous Groups (COG) and Eukaryotic Orthologous Groups (KOG) with e-value cutoff of 1e-5.

Differentially Expressed Gene (DEG) analysis

Gene expression was determined by the FPKM (Fragments Per kb of transcript per Million mapped fragments) method. The gene expression differences between the papilla and skin tissues were identified following the formula:Fold change = Log2FPKMPapillaFPKMSkin

DEGs were determined with the absolute fold change values greater than 2.0, and FDR (false discovery rate) lesser than 0.01 (Cui et al., 2014; Cui et al., 2015).

To further investigate DEGs identified between papilla and skin, genes were compared to those identified from the A. japonicus intestine RNA-Seq dataset (accession NO. GSE44995) from a previous study by Sun et al. (2013). The intestine is responsible for the metabolic rate depression under deep aestivating conditions (Chen et al., 2013) and plays a role in organ regeneration (Sun et al., 2013). In the present study, we used the intestine as a major site in the internal environment of A. japonicus to further investigate DEGs identified between papilla and skin. All assembled sequences of A. japonicus published in Sun et al. (2013) were downloaded as a database to blast the DEGs; the differential expression of DEGs among the papilla, skin, and intestine was estimated using the formula:Score=Log2FPKMTissuesRPKMIntestines

Where FPKMTissues indicates the FPKM of papilla or skin; RPKMIntestines indicates the RPKM of intestine. Significant candidates were determined as the absolute score greater than 4.0.

qRT-PCR validation

RNA-Seq results were validated by qRT-PCR analysis of 16 randomly selected DEGs. Primers were designed following the manufacturer’s recommendations of SYBR Premix Ex TaqTM II kit (Takara, Dalian, China). The β-actin was used as housekeeping. All the primers are shown in supplementary Table S1. Briefly, the amplification was performed in a total volume of 16 μL, containing 8 μL 2× SYBR Premix Ex Taq II, 1 μL of cDNA, and 0.3 μL of 10 μM of each gene-specific primer. The qRT-PCR reactions were performed on ABI stepone plus platform and replicated in three pools. And three technical replications were performed for each qRT-PCR validation. PCR was conducted as follows: 94 °C for 30 s, 45 cycles of 94 °C for 5 s, annealing temperature (showed in Table S1) for 15 s, and 72 °C for 15 s.

Gene enrichment analysis

The gene enrichment analysis was conducted using KEGG database. The over-presentation of the DEGs was determined in the specific pathways. The level of enrichment was indicated by enrichment factor, and p-value was used to calculate the significance of enrichment. The top 10 KEGG enrichments were selected to carry out further analysis.

Results

Sample sequencing

RNA-Seq of the papilla and skin samples yielded over 70 million pair-end reads with average length of 125 bp (Table 1). Similar number of reads was obtained from both tissues, with over 33 million reads from papilla and over 37 million from skin. After trimming, 30.6 and 36.4 million high-quality reads were retained from papilla and skin, respectively. Totally, 7.7 billion bases generated from the papilla, 9.2 billion bases generated for the skin and 8.6 billion bases generated for the tube foot were used for down-stream analysis of de novo assembly and mapping. Data obtained from papilla, skin and tube foot were deposited to the sequence read archive (SRA) with the accession numbers SRX1097860 and SRX1081978.

Table 1 Summary of the RNA-Seq data.

	Number of reads	Number of reads after trimming	Number of nucleotides after trimming (bp)	
Papilla	33,504,127	30,657,027	7,723,425,469	
Skin	37,384,685	36,444,908	9,182,161,309	
Total	70,888,812	67,101,935	16,905,586,778	

Transcriptome assembly and annotation

The de novo assembly resulted in a total of 156,501 transcripts, with the average length of 910.77 bp and N50 length of 1,694 bp (Table 2). The length distribution of transcripts and unigenes are shown in Fig. 1. Over 84% of reads from both tissues were successfully mapped back to the de novo transcriptome assembly.

Figure 1 The distribution of the size of transcripts and unigenes.

Length distribution of assembled transcripts (A) and unigenes (B) of sea cucumber (Apostichopus japonicus).

Table 2 Statistics of transcriptome reference assembly and annotation.

Assembly	Number of transcripts	156,501	
Maximum transcript length	18,781 bp	
Minimum transcript length	201 bp	
Average transcript length	910.77 bp	
N50 length	1,694 bp	
Number of mapped reads from the papilla	25,946,333 (84.6%)	
Number of mapped reads from the skin	30,913,283 (84.8%)	
Annotation	Unigenes with blast hits to NR	30,706	
Unigenes with blast hits to Pfam	22,261	
Unigenes with blast hits to Swiss-Prot	18,944	
Unigenes with blast hits to KOG	22,361	
Unigenes with blast hits to COG	10,876	
Unigenes with KEGG terms	11,190	
Unigenes with GO terms	12,140	
Total	33,584	

The transcriptome assembly was annotated by BLASTX against NCBI NR, Pfam, Swiss-Prot, KEGG, COG and KOG databases with E-value threshold of 1e-5. Annotation resulted in the identification of 92,343 unigenes (unique transcripts matched with known proteins). From all the 92,343 unigenes, 30,706 were found to have homologs in NR database, 22,261 found to posses functional domains in Pfam database; 18,944 unigenes showed significant matches to Swiss-Prot database, 22,361 to KOG, 11,190 to KEGG, 10,876 to COG and 12,410 unigenes were associated with GO terms (Table 2). Taken together, a total of 33,584 unigenes had at least one significant matches to these databases (Table 2). The unigenes annotated with NR database accounted for the largest proportion (91.4%), followed by Pfam and Swiss-Prot (Fig. 2).

Figure 2 The distribution of annotated unigenes across database.

Venn diagram display of the proportion of annotated unigenes in NR, Pfam, Swiss-Prot and GO.

Distribution of the 12,140 unique proteins in different GO categories is shown Fig. 3. The transcriptome was enriched in cell component GO categories related to cell part (22.8%) and cell (22.6%). For biological process, metabolic process (28.1%) was the most abundant GO categories. Regarding to molecular function, catalytic activity (45.5%) and binding (39.0%) were the most abundant GO categories. In the correlational study, Du et al. (2012) found that membrane-bounded organelle was the most represented GO term in cell component; the major category in biological process was the primary metabolic process; and genes involved in hydrolase activity accounted for major proportion in molecular function. To be noted, because the samples used in Du et al. (2012) study were collected from different developmental stages and adult tissues (intestines, respiratory trees and coelomic fluid), there may be some biases.

Figure 3 Distribution of the most common GO term categories.

Identification of DEGs

A total of 1,059 DEGs were identified between the papilla and skin. The MA plot showed significant DGE (blue) against all non-significant DEG (red) (Fig. 4A). Among identified DEGs, 739 were expressed at significantly higher in papilla, while 320 genes were expressed at significantly higher levels in skin (Table S2). The number of genes with higher expression levels in papilla was over twice than the number of that in skin. Papilla, as the projections of body wall, included more unique contents than skin, such as the calcareous ossicles, which are hidden in the dermis of body wall, papillae and tentacles (Steven et al., 2012). We also analysed the expression profiles of 1,059 DEGs in each tissue. Papilla-specific genes represent the DEGs that there is no expression in the skin, and that goes for skin-specific. A total of 288 papilla-specific DEGs were expressed only in papilla, while 171 DEGs were found to be only expressed in skin (skin-specific). A total of 600 DEGs were expressed in both papilla and skin (Fig. 4B). Apparently, the number of skin-specific (53.44%) genes is higher than papilla-specific genes (38.97%).

Figure 4 The DEGs in the papilla and skin of sea cucumber.

(A) M-A plots showing gene expression in papilla and skin. The x-axis represents the logarithm of FPKM and y-axis represents the logarithm of foldchange; (B) Venn diagram displays the number of papilla-specific, skin-specific, and co-expressed genes.

Of the 1,059 DEGs, 61 DEGs were annotated to homologous genes in strongylocentrotus purpuratus, a model species that is closely related to A. japonicus. Hsp gp96, Hsp26, ALDOA (aldolase class-1 protein) and tenasxin were annotated with A. japonicus. Our results revealed that Hsp gp96 and ALDOA were 3.93- and 4.45-Fold up-regulated in papillae, respectively. In constrast, Hsp26 and tenascin were −2.39- and −3.62-Fold down-regulated in skin, respectively. In addition, 236 differentially expressed genes were not annotated with any database, 160 of which were apparently higher in papilla. Further analysis revealed that 40 of which were papilla-specific and two were skin-specific.

Putative genes related to development that may be associated with the formation of the papilla were identified (Table 3). Detailed information of develop-related genes was provided in Table S3. Our results revealed that cuticle collagen 2 and alpha-2 collagen were highly expressed in papilla with 5.76 and 2.55, respectively. Several genes that know to be related to the collagen development (Hinz et al., 2003; Hinz, 2009; Leask & Abraham, 2004), such as Fibroblast Growth Factor (FGF), Transforming Growth Factor-β (TGF-β) and Integrin-α2 (ITGA2) were found to be significantly expressed. Several Ras-related genes such as Ran, Rab1a, Arf3, Ran1, Ras, RhoA, Rho Guanine nucleotide exchange factors (RhoGEF), Rho GTPase, Rho GTPase activation protein (RhoGAP) and Ran-binding protein 1 (RanBP1), which play key roles in the development by regulating growth and morphogenesis, were also identified in our study (Table S4). All Ras-related DEGs were expressed at lower levels in skin except for RhoGEFs that were reported to be associated with cancer, pathogen infection or neural system related diseases and development (Reichman et al., 2015). Understanding of the function of Ras-related genes will facilitate to unravel the mechanisms of some physiological and pathological process in the skin of A. japonicus.

Table 3 Differentially expressed genes between the papilla and skin that are involved in development.

Unigene ID	Gene symbol	Foldchange	
c14695.graph_c0	cdk	4.96	
c40875.graph_c0	cyc-B	6.26	
c75877.graph_c0	cyc-A	6.346	
c76406.graph_c0	cytC	2.76	
c76859.graph_c0	gadd45a	4.82	
c12901.graph_c0	ck2bl	4.40	
c14611.graph_c0	MAGUKs	4.34	
c18023.graph_c0	PP2A	4.54	
c19039.graph_c0	claudin	6.31	
c37832.graph_c0	actin	6.05	
c16255.graph_c0	FGF	5.26	
c58770.graph_c0	ITGA2	2.12	
c15897.graph_c0	TGF-β	6.21	
c54738.graph_c0	col-α2	2.55	
c54237.graph_c0	tub-α	−4.53	
c77661.graph_c0	emmhc	4.49	
c54933.graph_c0	eef2	−5.43	
c42633.graph_c0	gtf 8	4.96	
c57892.graph_c0	ubeE2	−4.18	
c76626.graph_c0	ctATPase	4.62	
c38162.graph_c0	cul-α2	5.76	

To further verify DEGs data, we compared our results with those DEGs identified in the intestine of A. japonicus from a previous study Sun et al. (2013) (results are shown in Table 4). Seven DEGs showed the same score trend as that of the fold change in papilla. The reason for this observation could be due to the lack of a complete RPKM data (Sun et al., 2013).

Table 4 The result of DEGs with significantly different expression by comparison with the intestine.

Unigene ID	Isotig ID	Annotate	Foldchange	Score	
c45050.graph_c0	isotig25664	Cell death abnormality protein 1	8.62	14.40	
c64723.graph_c0	isotig15743	Sushi domain (SCR repeat)	7.15	4.09	
c67657.graph_c0	isotig19241	Fibrinogen-like protein A	3.02	4.96	
c66534.graph_c0	isotig15670	Hypothetical protein CAPTEDRAFT_211426	2.84	11.35	
c73725.graph_c2	isotig27287	Sulfotransferase family	2.56	12.76	
c60095.graph_c0	isotig09563	Hypothetical protein BRAFLDRAFT_231341	2.54	4.31	
c60588.graph_c0	isotig18328	Histone-lysine N-methyltransferase	2.16	13.36	
Note:

The “Isotig ID” column indicates the gene ID from the data of Sun et al. (2013).

qRT-PCR validation

To validate the DEGs results obtained, we randomly selected 16 DEGs for validation using qRT-PCR. As shown in Fig. 5, the DGEs identified from qRT-PCR analysis were correlated well with those obtained from qRT-PCR, indicating the reliability and accuracy of the RNA-Seq method used in the present study.

Figure 5 Comparison between RNA-Seq results and qRT-PCR validation results.

X-axis shows genes in two tissues validated in this study; Y-axis shows Log2 Ratio of expression of SK (skin) versus YZ (papilla). AAC4PL, AAC-rich mRNA clone AAC4 protein-like; Hsp26, heat shock protein 26; NP, novel protein; TN, Tenascin; EMI, EMI domain; Hp TTRE, hypothetical protein TTRE_0000953901; FGL, Fibrinogen-like protein A; HpX975-24482, hypothetical protein X975_24482, partial; PP2A, Serine/threonine-protein phosphatase; FIL2L, Fibrinogen-like protein A; Col-α FiCollagen gen-like protein A; Col-l-ase; FIL2L, Fibrinogen-like proteiphosphatealdolase; ITGA2, Integrin alpha 2; MAD2A, Mitotic spindle assembly checkpoint protein.

Enrichment analysis of DEGs

A total of 296 DEGs identified were mapped to 133 pathways. KEGG enrichment pathway analysis was also carried out to investigate their potential functional roles. The top 10 enrichment pathways were selected by a hypergeometric test (p < 0.05) (Table 5). One of which is the ribosome pathway, which was related to the protein biogenesis and was observed to be involved in intestine regeneration (Sun et al., 2013) and aestivation (Chen et al., 2013; Zhao et al., 2014a; Zhao et al., 2014b) in the A. japonicus. In addition, tight junction and p53 signaling pathways were also detected in enrichment pathways analysis (detailed information is provided in Table S5).

Table 5 Enrichment analysis of genes with significantly differential expression between the papilla and skin.

Pathway	KO	Enrichment factor	p-value	
Ribosome	ko03010	0.39	1.47E-13	
Oocyte meiosis	ko04114	0.27	2.07E-06	
Cell cycle	ko04110	0.40	0.000557178	
Glycolysis/Gluconeogenesis	ko00010	0.43	0.000610528	
p53 signaling pathway	ko04115	0.28	0.003222732	
Tight junction	ko04530	0.38	0.004679565	
NOD-like receptor signaling pathway	ko04621	0.27	0.010624351	
Regulation of actin cytoskeleton	ko04810	0.44	0.017102208	
RNA transport	ko03013	0.68	0.029611824	
Progesterone-mediated oocyte maturation	ko04914	0.52	0.033017313	

Discussion

In this study, we conducted comparative transcriptome analysis between papilla and skin, two important organs of sea cucumber. A total of 1,059 differentially expressed genes were identified between the two organs. This result lay the foundation to identify genes that were potentially involved in the development of the papilla and skin. The generated genomic resources should be valuable for other genetic and genomic studies in the A. japonicus.

As previously reported, excessive deposition resulting from abnormal balance of growth factors and cell proliferation can improve local hyperplastic collagen production in skin in response to injury in mammalians (Tuan & Nichter, 1998). Keloids (Seifert & Mrowietz, 2009; Shih & Bayat, 2010) and Hypertrophic Scar (HS) (O’Leary, Wood & Guillou, 2002), are characterized by fibroblastic proliferation and accumulation of Extracellular Matrix (ECM), especially excessive deposition of collagen. However, such prominences are regarded as benign tumors (Diao et al., 2011). It has been suggested that factors such as FGF, ITGA2, TGF and S-adenosylmethionine (a-SMA) can cause those lesions (Hinz et al., 2003; Hinz, 2009; Leask & Abraham, 2004). The FGF activity was first identified from bovine pituitary in 1974 (Gospodarowicz & Moran, 1974). FGF signaling is required for different developmental stages during embryogenesis (Sun et al., 1999; Naiche, Holder & Lewandoski, 2011; Niwa et al., 2011; Vega-Hernández et al., 2011). Compared with normal dermal fibroblast, TGF-β is believed to induce collagen production and increase the contractile activity in keloid fibroblasts (Bran et al., 2010; Sandulache, Parekh & Li-Korotky, 2007). In additon, TGF-β associated with connective tissue growth factor (CCN2) has been revealed to stimulate a-SMA, collagen expression (Jiang et al., 2008). ITGA2 is the main cell adhesion molecule that takes part in the modulation of collagen contraction and the activity of myofibroblast in HS (Cooke, Sakai & Mosher, 2000). The expression levels of ITGA2 were also found up-regulated in hypertrophic scar fibroblasts, compared with normal skin tissues in human. In our study, we found that col-α2 was expressed at a higher level in the papilla, and we also observed differential expression patterns of genes involved in collagen synthesis as the major differences between the papilla and skin. The expression of FGF, TGF-β and ITGA2, associated with collagen development, were all expressed at higher levels in the papilla. Compare to previous studies of local hyperplastic collagen in mammals, we speculate that these collagen-related genes may play critical roles in stimulating the production of collagen in papilla and might be involve in the morphological differentiation between the two organs.

Besides the development-related genes as discussed above, some immune-related genes are also identified as being differentially expressed between papilla and skin in this study, such as Hspgp96, Hsp26, ALDOA and tenascin. Many lines of evidences support that Hsps act as natural immunoregulatory agents, increasing the awareness of innate immune cells to pathogens (Ciancio & Chang, 2008; Prohaszka & Fust, 2004; Zugel & Kaufmann, 1999). ALDOA plays a role in glycolysis pathway (Oparina et al., 2013). Further investigation is required to explore the pathological researches for papilla and skin. Through analysis of the DEGs against intestine transcriptome data from a previous study (Sun et al., 2013), we found Fibrinogen-like protein A(fglA) showed the same score trend as that of fold change in papilla. FglA is a member of the fibrinogen-related protein superfamily, plays crucial roles including innate immune response, regeneration and blood clotting (Yamamoto et al., 1993). Previous studies have demonstrated that fglA is widely distributed in A. japonicus body wall, intestines, longitudinal muscles and respiratory tree of A. japonicus (Wu et al., 2014). Our results also show that fglA was 3.02 fold change and 4.96 score up-regulated in papilla, respectively. The role of fglA in the development of papilla remains unclear and further investigation is required to understand its functional roles.

Enrichment KEGG analysis revealed that tight junction and p53 signaling pathway were highlighted in enrichment pathways. In humans, the content of G2-M arrested cells in keloid skin was higher than in normal skin (Varmeh et al., 2011). Keloid fibroblasts showed a higher rate of senescence and lower proliferative capacity in comparison to normal fibroblasts (Varmeh et al., 2011). In our study, a set of genes, including growth arrest and DNA-damage-inducible protein (gadd45), cyclin dependent kinase 1 (cdk), cyclin-B (cyc-B), cyclin-A (cyc-A) and cytochrome C (cyt-C), were all expressed at higher levels in the papilla (Table 1). These genes are involved in the p53 signaling pathway. Once p53 signaling pathway is activated, it can induce either cell cycle arrest or apoptosis in the damaged cell. In humans, cyc-B and cdk2 kinase influence a cell’s progress through the cell cycle, which is especially important in several skin cancers (Ely et al., 2005; Casimiro et al., 2014). Cyc-B forms the regulatory subunits and cdk2 form the catalytic subunits of an activated heterodimer. The cyc-B has no catalytic activity, and cdk2 is inactive in the absence of a partner cyc-B. Once activated the cdk2/cyc-B complex control cell cycle (Abreu Velez & Howard, 2015). Gadd45 is a ubiquitously expressed 21 protein with a key role in response to genotoxic agents, and it is involved in many biological processes related to maintenance of genomic stability and apoptosis. It has been shown that gadd45’s inhibits cdk2 kinase activity through alteration of cyc-B subcellular localization, inducing the arrest of the cell cycle in G2-M state (Jin et al., 2000; Smith et al., 1994). These results indicated that the level of cell cycle arrest at the G2-M in the papilla might be higher than in the skin. It’s speculated that papilla fibroblasts commit to a higher rate of senescence, which may cause fibroblast-related genes eventually stop expressing and maintain external morphology of the papilla.

Genes involved in tight junction were enriched in papilla. Tight junctions are essential for epithelial morphology, which can form seals between epithelial cells and create a selectively permeable barrier to intercellular diffusion (Zheng et al., 2011). Besides, we also found that the expression of Serine/threonine-protein phosphatase (PP2A) in papilla is higher than that in the skin. Many reports showed that PP2A regulates Ataxia Telangiectasia Mutated (ATM), Ataxia Telangiectasia Rad3 related (ATR), Check Point Kinase-1 (CHK1), and Checkpoint Kinase-2 (CHK2) after DNA damage, and activate the checkpoint of G2-Massociated with the p53 signaling pathway. The process activated by PP2A may also regulate the external morphological of papilla and skin of A. japonicus.

Conclusion

In this study, we performed comparative transcriptome analysis of the skin and papailla A. japonicus by using RNA-Seq. In total, 156,501 transcripts and 92,343 unigenes were assembled. A total of 1,059 differentially expressed genes were indentified between the two important organs of A. japonicus. We identified 236 novel genes (not annotated with any database), 160 of which were expressed at higher levels in papilla. Further tissue-expression analysis identified 288 papilla-specific genes and 171 skin-specific genes. Gene pathway enrichment analysis revealed several gene pathways that were involved in development. In addition, many DEGs involved in the process of p53 signaling pathway and tight junction were also identified, which were ported to be relative to keloid skin in humans. This result provided insight into genes and pathways that may be associated with the formation of the papilla and skin in sea cucumber, laying the foundation for further investigation to understand the development of the papilla in A. japonicus. Moreover, the generation of larger-scale transcriptomic data presented in this work enriched genetic resources of the echinodermata species, which should be valuable to comparative and evolutionary studies in echinoderms.

Supplemental Information

Supplemental Information 1 PCR primers used for qRT-PCR validation.

Click here for additional data file.

Supplemental Information 2 Differentially expressed genes identified by transcriptome comparison between papilla and skin.

Click here for additional data file.

Supplemental Information 3 The detailed information of putative genes related to development.

Click here for additional data file.

Supplemental Information 4 Identification of Ras-related genes in the DEGs.

Click here for additional data file.

Supplemental Information 5 The detailed information of KEGG enrichment pathways.

Click here for additional data file.

We thank Dr. Daniel Garcia de la serrana (Fish Muscle Research Group, Scottish Oceans Institute, University of St Andrews, UK) for revising and polishing the manuscript. We would like to acknowledge Prof. Tzi Bun Ng and two anonymous reviewers for helpful comments on the manuscript.

Additional Information and Declarations

Competing Interests

Author Contributions

DNA Deposition

The authors declare that they have no competing interests.

Xiaoxu Zhou conceived and designed the experiments, performed the experiments, analyzed the data, contributed reagents/materials/analysis tools, wrote the paper, prepared figures and/or tables, reviewed drafts of the paper.

Jun Cui analyzed the data, wrote the paper, prepared figures and/or tables, reviewed drafts of the paper.

Shikai Liu analyzed the data, contributed reagents/materials/analysis tools, wrote the paper, prepared figures and/or tables, reviewed drafts of the paper.

Derong Kong performed the experiments, reviewed drafts of the paper.

He Sun performed the experiments, reviewed drafts of the paper.

Chenlei Gu performed the experiments, reviewed drafts of the paper.

Hongdi Wang contributed reagents/materials/analysis tools, reviewed drafts of the paper.

Xuemei Qiu contributed reagents/materials/analysis tools, reviewed drafts of the paper.

Yaqing Chang contributed reagents/materials/analysis tools, reviewed drafts of the paper.

Zhanjiang Liu wrote the paper, reviewed drafts of the paper.

Xiuli Wang conceived and designed the experiments, analyzed the data, contributed reagents/materials/analysis tools, wrote the paper, reviewed drafts of the paper.

The following information was supplied regarding the deposition of DNA sequences:

1. Raw reads of papilla deposited in sequence read archive database (SRX1097860).

2. Raw reads of skin deposited in sequence read archive database (SRX1097860).

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
