# Peer review of "Comparative transcriptome analysis of papilla and skin in the sea cucumber, Apostichopus japonicus"

_PeerJ, doi:10.7717/peerj.1779_

## Round 0.1 · original submission · Major Revisions

Dear authors

Please take into account all of the reviewers' comments when revising your manuscript.

Merry Christmas & happy New Year
TB Ng

Reviewer 1 ·

Basic reporting

1. Please confirm whether the official symbol of integrin-alpha 2 is ‘IN-alpha2’ or not.

Experimental design

1. This manuscript is focused on determination of global changes in gene expression between the papilla and skin in the sea cucumber, however, there is no information about how to prepare the papilla and skin tissues (samples) from the see cucumber in Materials and Methods section.
2. Also, the authors further investigated the DEGs (differentially expressed genes) between the papilla and skin in the see cucumber by comparing to see cucumber intestine data from Sun et al. (Sun et al 2013). However, the purpose of this comparing analysis and the meaning of the results were little described in the Materials-and-Methods and Discussion section. Also, please clarify the meaning of the following statement; “Only 7 of DEGs identified showed the same foldchange trend than observed in the papilla.”. Furthermore, the sentence of “Fibrinogen-like protein A (FGLA) … and blood clotting” is should be supported by inserting several references. Also, the authors only described the result of commonly dysregulated genes between DEGs in the papilla and the previous reported intestine data by Sun et al. in the Result section and Table 4. Where is results from commonly dysregulated genes between DEGs in the Skin and the intestine data? And, “Foldchang” should be corrected to “Fold change” in Table 4.

Validity of the findings

1. The authors confirmed and validated the DEGs results by performing qRT-PCR experiment in Figure 5, however, there is no error bars in the result. The results of qRT-PCR data was obtained from single experiments, not triplicate experiments?

Additional comments

The word spacing in the manuscript should be confirmed more detailed.

Reviewer 2 ·

Basic reporting

The data is huge and potentially very informative for sea cucumber researchers. This article will be worth to publish in PeerJ after extensive revision.

Experimental design

Sound experimental techniques.

Validity of the findings

Poor organization of the data. Please see my statement that indicated below.

Additional comments

This article was written with poor organization that makes readers hard to follow the points the authors want to communicate.
By following the title of this article, the text should categorically described, such as (1) Papilla, (2) Skin and (3) Tube feet, and so on. Better not chaotically hopping around little by little among organs. Should be much more focused in each section. I think Fig 3 and Fig. 4B are the most valuable information for the readers. Thus, text better be more precise on each subject. Others are mere technical descriptions employed in the present study. These latter subjects should be as concise as possible, unless some of them involve something new.
The followings are my suggestion of practical way to revise the text.
(1) “Abstract” lacked focused description indicated by Fig. 3 and Fig. 4B.
(2) “Introduction” described wordy aimless background. The authors should enrich the statement represented by line 94-100.
(3) “Results” should re-organize categorically by focusing on skin and papilla as I pointed above. Other organs or tissues should be presented as an additional section.
(4) Line 229-235 better move to “Introduction”.
(5) Typing style was truncated.

The minor revisions are the follows.
(1) Line 191 to 193: Lack of necessity in logic. Better be deleted.
(2) Line 210 to 212: Move to Discussion.
(3) Line 229-235: Move to Introduction.
(4) “Conclusion” (Line 298-307): The current description is “Future perspective”. Needs complete re-writing by focusing on the main results of this study.
I would recommend professional editing work before submitting the revised version.
I have a good sample article that subjected similar study but on sea urchin genome project. Following is the information. I welcome the authors to refer the way to re-write.
They are in Developmental Biology volume 300 Issue 1 “Sea Urchin Genome: Implications and Insight” 2006. I was not involved in any of articles in the issue.

---

## Round 0.2 · accepted · Accept

Thank you for your contribution to PeerJ, it is now Acceptable